# Experimental Lagos bat virus infection in straw-colored fruit bats: A suitable model for bat rabies in a natural reservoir species

Lineke Begeman[1☉]*, Richard Suu-Ire[2,3☉], Ashley C. Banyard[4], Christian Drosten[5], Elisa Eggerbauer[6,7], Conrad M. Freuling[6], Louise Gibson[3], Hooman Goharriz[4], Daniel L. Horton[8], Daisy Jennings[4], Denise A. Marston[4], Yaa Ntiamoa-Baidu[9], Silke Riesle Sbarbaro[3,10], David Selden[4], Emma L. Wise[4], Thijs Kuiken[1], Anthony R. Fooks[4], Thomas Müller[6], James L. N. Wood[10], Andrew A. Cunningham[3]*

1 Department of Viroscience, Erasmus University Medical Centre, Rotterdam, The Netherlands, 2 School of Veterinary Medicine, College of Basic and Applied Sciences, University of Ghana, Accra, Ghana, 3 Institute of Zoology, Zoological Society of London, Regent's Park, London, United Kingdom, 4 Wildlife Zoonoses and Vector Borne Disease Research Group, Animal and Plant Health Agency, Addlestone, United Kingdom, 5 Institute of Virology, Medical University of Berlin, Berlin, Germany, 6 Institute of Molecular Virology and Cell Biology, Friedrich-Loeffler-Institut, Federal Research Institute for Animal Health, Greifswald, Island of Riems, Germany, 7 Thüringer Landesamt für Verbraucherschutz, Bad Langensalza, Thüringen, Germany, 8 School of Veterinary Medicine, University of Surrey, Guildford, United Kingdom, 9 Centre for African Wetlands / Department of Animal Biology and Conservation Science, University of Ghana, Accra, Ghana, 10 University of Cambridge, Cambridge, United Kingdom

☉ These authors contributed equally to this work.
* l.begeman@erasmusmc.nl (LB); a.cunningham@ioz.ac.uk (AAC)

**Data Availability Statement:** All relevant data are within the manuscript and its Supporting Information files.

## Abstract

Rabies is a fatal neurologic disease caused by lyssavirus infection. Bats are important natural reservoir hosts of various lyssaviruses that can be transmitted to people. The epidemiology and pathogenesis of rabies in bats are poorly understood, making it difficult to prevent zoonotic transmission. To further our understanding of lyssavirus pathogenesis in a natural bat host, an experimental model using straw-colored fruit bats (*Eidolon helvum*) and Lagos bat virus, an endemic lyssavirus in this species, was developed. To determine the lowest viral dose resulting in 100% productive infection, bats in five groups (four bats per group) were inoculated intramuscularly with one of five doses, ranging from $10^{0.1}$ to $10^{4.1}$ median tissue culture infectious dose ($TCID_{50}$). More bats died due to the development of rabies after the middle dose ($10^{2.1}$ $TCID_{50}$, 4/4 bats) than after lower ($10^{1.1}$, 2/4; $10^{1.1}$, 2/4) or higher ($10^{3.1}$, 2/4; $10^{4.1}$, 2/4) doses of virus. In the two highest dose groups, 4/8 bats developed rabies. Of those bats that remained healthy 3/4 bats seroconverted, suggesting that high antigen loads can trigger a strong immune response that abrogates a productive infection. In contrast, in the two lowest dose groups, 3/8 bats developed rabies, 1/8 remained healthy and seroconverted and 4/8 bats remained healthy and did not seroconvert, suggesting these doses are too low to reliably induce infection. The main lesion in all clinically affected bats was meningoencephalitis associated with lyssavirus-positive neurons. Lyssavirus antigen was detected in tongue epithelium (5/11 infected bats) rather than in salivary gland epithelium (0/11), suggesting viral excretion via the tongue. Thus, intramuscular inoculation of $10^{2.1}$ $TCID_{50}$ of Lagos bat virus into straw-colored fruit bats is a suitable model for

**Funding:** This study was financially supported by the European Union FP7-funded project Anticipating the Global Onset of Novel Epidemics (ANTIGONE), project number 278978 to TK, AAC, ARF, JLNW and CD, and by the UK Department for Environment, Food and Rural Affairs (Defra), Scottish and Welsh Government by grant SE0426 to ARF and ACB. AAC was supported by a Royal Society Wolfson Research Merit award. The funders had no role in study design, data collection and analysis, decision to publish, or preparation of the manuscript.

**Competing interests:** The authors have declared that no competing interests exist.

lyssavirus associated bat rabies in a natural reservoir host, and can help with the investigation of lyssavirus infection dynamics in bats.

## Author summary

Rabies is a fatal neurologic disease affecting people and animals. Rabies is caused by infection with a virus of the genus Lyssavirus. People usually get infected from dog bites, but bats are an increasingly important source of the disease. To better understand the biology of rabies in bats, we developed a laboratory model to study the disease in bats under controlled circumstances. For this model we used Lagos bat virus in straw-colored fruit bats and, as part of its development, we wanted to know the best virus dose to use to cause rabies. Therefore, we compared the outcomes of five different virus doses injected into the muscle of the bats. The best dose for our model was the middle dose, which caused rabies more frequently than either the highest or the lowest doses. The higher doses more frequently resulted in the development of an anti-viral immune response which appeared to protect against disease, while bats with low doses also often failed to develop disease. The virus dose thus followed the Goldilocks principle, with the middle dose being just right.

## Introduction

Rabies is an almost invariably fatal disease caused by rabies virus (RABV) or any other member of the *Lyssavirus* genus (family: *Rhabdoviridae*, order: Mononegavirales) [1,2]. Rabies virus is predominantly transmitted to people by carnivores, in particular the domestic dog (*Canis familiaris*), and causes more than 59,000 human fatalities annually [3]. As terrestrial rabies in domestic and wild carnivores is being brought under control by vaccination in high- and middle-income countries, the role of bats as a source of human infection has become more evident [4,5]. Also, bats are the main source of cattle rabies infections in South America [6]. In Latin America, an estimated 30 million US$ is spent annually on rabies preventive measures [7]. Although rare, the transmission of RABV from bats to terrestrial carnivores has been demonstrated as a driver of outbreaks in terrestrial mammals, as has been reported several times in the Americas [8–13]. Singular spill-over events of lyssaviruses other than RABV from bats to terrestrial mammals also have been reported [14–19]. Despite increasing recognition of their importance, relatively little is known about the dynamics of lyssavirus infections in their natural hosts, bats [13,20,21]. Based on the recurrent finding of high seroprevalences in some free-ranging bat populations (Table 3.2 in [22]), a number of researchers have hypothesized that bats, in contrast to other mammals, can survive a productive lyssavirus infection, i.e an infection where lyssavirus reaches the brain and is subsequently excreted from the oral cavity [23]. In contradiction, there is no indication from experimental infections in bats that this occurs [24–47] and bats seem, like other species, unable to survive infection of the brain, which appears to always result in fatal rabies [21,48]. Experimental infections in bats have been performed mainly with four lyssaviruses: RABV, Australian bat lyssavirus, and European bat lyssaviruses 1 and 2 [49]. These experimental infections have shown that—similar to RABV in carnivores—these lyssaviruses target the brain [50–52] with infection typically leading to encephalitis and death [26,32,35,37]. An important limiting factor in this previous bat-lyssavirus research is that oral excretion of virus was rarely observed in experimentally-infected bats making it difficult to test the hypothesis that bats are able to survive a productive

lyssavirus infection [24–46]. The rare observation of oral excretion suggests experimental models for natural lyssavirus infections in bats need to be improved.

The current study is a second step in the development of an experimental model that meets our goal of mimicking natural infection in a natural reservoir bat host. By 'natural reservoir host' we mean a host that is naturally infected and has co-evolved with the pathogen. Lagos bat virus (LBV), which comprises four lineages, A to D [49], is endemic in the straw-colored fruit bat (*Eidolon helvum*) [53,54], a common and widespread bat species in sub-Saharan Africa, which is not considered as 'Threatened' by the International Union for Conservation of Nature (www.iucnredlist.org). In a previous study, we tested different Lagos bat virus isolates in the straw-colored fruit bat [55]. Based on that study, a recent LBV isolate from Ghana [56] was selected as most appropriate because viral infection spread from the brain widely in the peripheral nervous system as expected for a natural productive lyssavirus infection [55].

Infections of LBV in mammals other than bats have been reported sporadically [17,18,57]. Human infection has not been demonstrated, but diagnostic analysis of human rabies cases in Africa, if undertaken, typically uses methods that do not distinguish RABV from LBV or other lyssaviruses [49]. While the impact of LBV on human health is unknown, the widespread distribution of the straw-colored fruit bat and the apparent high rate of exposure of this species to LBV across its range [54,58], the increasingly close association of people with this gregarious bat species which often roosts in large human settlements and is consumed as food [59] and the failure of rabies immunization to protect against LBV [60], indicate that this pathogen has the potential to threaten public health.

The specific goal of the current study was to identify the lowest inoculation dose of LBV that leads to a 100% rate of infection in straw-colored fruit bats after intramuscular inoculation. Additionally, the pathogenesis of LBV infection in these bats was investigated. The ultimate aim of this study was to develop a model that mimics a natural infection. Therefore, virus distribution, cell tropism and the lesions that developed in these experimentally-infected bats were compared with those of a naturally-infected bat, which also was the source of the LBV isolate we used for experimental inoculation [56].

## Results

### Clinical signs

Overall, 11 of 20 experimentally-inoculated bats died due to LBV infection and from now on these bats are referred to as having been rabid, whether or not clinical rabies was observed. Disease causation was confirmed as being LBV infection by positive reverse transcription quantitative PCR (RT-qPCR), rabies tissue culture infection test (RTCIT), fluorescent antibody test (FAT), and immunohistochemical (IHC) examination of their brains (see below). The $10^{2.1}$ medium tissue culture infectious dose (TCID$_{50}$) virus dose was the most successful in causing lethal LBV infection (Fig 1, Table 1).

Day of death post inoculation (pi) due to LBV infection ranged from 7 to 17, with one outlier (bat 14) from the $10^{0.1}$ TCID$_{50}$ group, that died on day 61 pi (Fig 1). Four bats died without clinical signs being observed, one each from the $10^{0.1}$ TCID$_{50}$, $10^{1.1}$ TCID$_{50}$, $10^{2.1}$ TCID$_{50}$ and $10^{4.1}$ TCID$_{50}$ groups, suggesting a short disease course. Seven bats developed clinical signs, of which six died or were euthanized within 12 hours, and one within 24 hours, of clinical signs first being observed. Clinical signs observed were increased vocalization, muscle spasms and tremors, increased saliva surrounding the mouth and aggression, and were the same as those noted in bats inoculated intracranially with this strain of LBV in a previous experiment [55]. There was no correlation between age (S1 Table) and infection status ($\chi^2$ [1, $N$ = 20] = 0.90, $p$ = .34), or between age and the development of antibodies ($\chi^2$ [1, $N$ = 20] = 1.62, $p$ = .20, an alpha level of .05 was used).

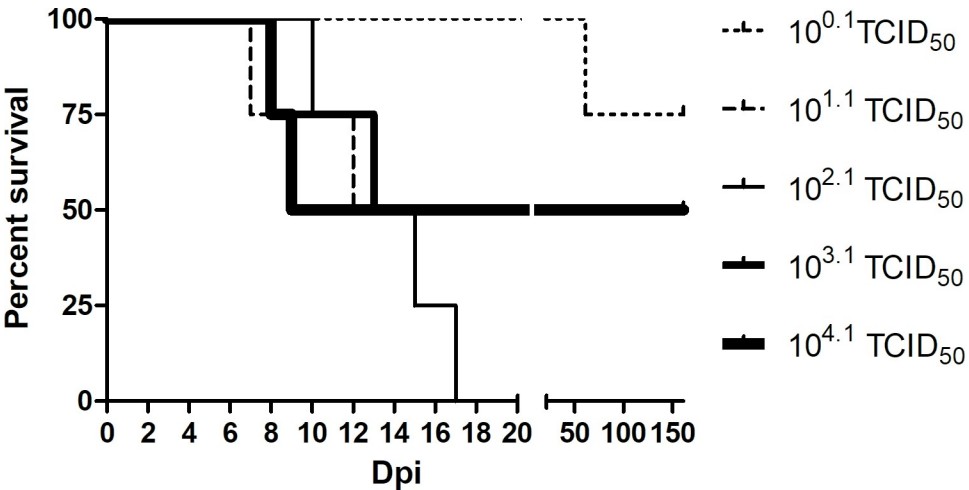

**Fig 1. Survival per inoculation group (n = 4 bats) over time, showing 100% mortality in the middle dose ($10^{2.1}$ TCID$_{50}$) group.**

**Table 1. Association between clinical outcome and detection of lyssavirus in the brains of straw-colored fruit bats inoculated with different doses of Lagos bat virus.**

| Virus dose group (TCID$_{50}$) | Bat no. | Clinical signs observed? | DoD pi[b] | Outcome of lyssavirus detection in brain by different methods | | | |
|---|---|---|---|---|---|---|---|
| | | | | RT-qPCR | RTCIT | FAT | IHC |
| $10^{0.1}$ | 17 | No | na[c] | - | - | - | - |
| | 18 | No | na | - | - | - | - |
| | 19 | No[a] | 61 | + | + | + | + |
| | 20 | No | na | - | - | - | - |
| $10^{1.1}$ | 13 | Yes | 12 | + | + | + | + |
| | 14 | No[a] | 7 | + | + | + | + |
| | 15 | No | na | - | - | - | - |
| | 16 | No | na | - | - | - | - |
| $10^{2.1}$ | 9 | Yes | 10 | + | + | + | + |
| | 10 | Yes | 13 | + | + | + | + |
| | 11 | Yes | 17 | + | + | + | + |
| | 12 | No[a] | 15 | + | + | + | + |
| $10^{3.1}$ | 5 | No | na | - | - | - | - |
| | 6 | No | na | - | - | - | - |
| | 7 | Yes | 8 | + | + | + | + |
| | 8 | Yes | 13 | + | + | + | + |
| $10^{4.1}$ | 1 | No[a] | 8 | + | + | + | + |
| | 2 | No | na | - | - | - | - |
| | 3 | No | na | - | - | - | - |
| | 4 | Yes | 9 | + | + | + | + |

[a] Bat was found dead, clinical signs were not observed

[b] Number of days between date of inoculation and date of death (DoD) or euthanasia because of clinical signs.

[c] Not applicable: bat remained healthy and was euthanized at the end of the experiment at day 160 or 161 post inoculation.

No clinical signs were reported in the naturally-infected bat that was the host for the virus used within this study, although this animal was not observed for long as it was killed at the time of capture [56].

## Serology

Four of the six experimentally-inoculated bats that developed LBV-neutralizing antibodies in their serum were in the two highest virus dose groups ($10^{3.1}$ and $10^{4.1}$ TCID$_{50}$), and none of these six bats were in the lowest virus dose group ($10^{0.1}$ TCID$_{50}$) (Fig 2). The development of antibodies thus seemed to be correlated with high virus dose, although the small number of bats per virus dose group precludes meaningful statistical analysis. The earliest that antibodies were detected in any bat was at 9 days pi.

Of the six bats that developed antibodies, four survived and remained clinically healthy to the end of the study (160 days post inoculation). The two seropositive bats that died, did so with clinical signs of rabies. However, the highest reciprocal antibody titer measured in these two bats was 27, which was lower than the highest titer measured in most of the bats that sero-converted and survived (Fig 2).

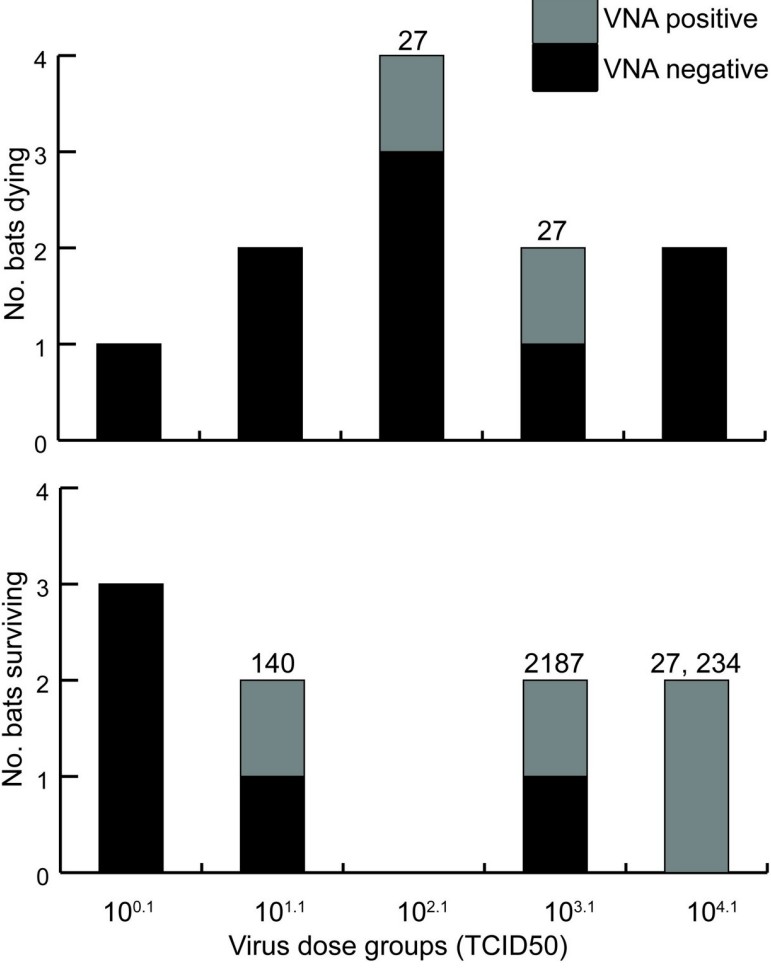

**Fig 2. Comparison of virus-neutralizing antibody response between straw-colored fruit bats that died or survived after inoculation with different doses of Lagos bat virus.** The numbers on top of the stacked column are reciprocal VNA titers.

In the naturally-infected bat used in this study no LBV-neutralizing antibodies were detected in serum using the modified rapid fluorescent focus inhibition test [56].

## Virology

Several parts of the brain (rostral part, hippocampus, cerebellum, medulla oblongata) of each experimentally-inoculated bat that died without clinical disease being observed or that had clinical signs and was euthanized (n = 11), tested positive for LBV using RT-qPCR, RTCIT, and FAT, while the brains of bats that survived until the end of the experiment (n = 9) tested negative (Table 1). An analysis of variance (one-way ANOVA) showed that the (log transformed) Ct values, reflecting the quantity of virus present, of the four different brain areas did not differ significantly, $F(1,62) = 5.94$, $p = .200$ (an alpha level of .05 was used). The mean brain Ct value per LBV-positive bat ranged from 26 to 28 in all but one bat, in which the mean Ct value was 30 (Table 2). These data suggest that when the virus reached a certain threshold in the brain, the bat either died after a short disease duration, or developed end-stage clinical disease and was euthanized.

Extra-encephalic tissues in 8 of the 11 bats (bats 19 [$10^{0.1}$ TCID$_{50}$ group], 13, 14 [$10^{1.1}$ TCID$_{50}$ group], 9, 10, 12 [$10^{2.1}$ TCID$_{50}$ group], 7 [$10^{3.1}$ TCID$_{50}$ group], and 1 [$10^{0.1}$ TCID$_{50}$ group]) that died with LBV infection tested positive by RT-qPCR. The tissues that tested positive by RT-qPCR were tongue and salivary gland (Table 3), liver (bat 19, Ct 39, and bat 13, Ct 32), lung (bat 19, Ct 38) and intestines (jejunum of bat 19, Ct 40; colon of bat 19, Ct 39; and duodenum of bat 13, Ct 38). Virus isolation confirmed the presence of LBV in all these tissues, apart from the tongue of bat 8, livers of bat 13 and 19, and the duodenum of bat 13, from which virus could not be isolated. From bats that survived to the end of the study all extra-encephalic tissues tested negative by RT-qPCR. All oral swabs taken from bats that died during the study were tested, as well as the oral swabs taken on the day of termination of the study from healthy bats that survived to the end of the study. All oral swabs in RNAlater (n = 97) tested negative for the presence of lyssavirus RNA using RT-qPCR, and 14 (14%) tested negative for beta actin. MEM stored oral swabs (n = 81) tested negative for the presence infectious virus by RTCIT. The FAT was performed for salivary gland tissue, and was inconclusive because of nonspecific fluorescence, as has been described previously for glandular tissue [61].

The brain of the naturally-infected bat had tested positive for LBV using a generic RT-PCR and FAT [56].

## Pathology

At necropsy, gross lesions were noted in 3 (nos. 12, 13, and 14) of the 20 bats from the experiment. All three were bats that died of rabies. Bat 13 and 14 had acute incision wounds on their

**Table 2. Comparison of survival time and relative quantity of Lagos bat virus RNA in the brains of straw-colored fruit bats inoculated with different doses of Lagos bat virus.** (Four bats in each virus dose group.)

| Virus dose group (TCID$_{50}$) | No. bats with fatal LBV infection | Survival time (range, days post inoculation) | Mean relative quantity of lyssaviral RNA in brain, per bat (Ct)[a] |
|---|---|---|---|
| $10^{0.1}$ | 1 | 61 | 26 |
| $10^{1.1}$ | 2 | 7–12 | 26, 26 |
| $10^{2.1}$ | 4 | 10–17 | 27, 28, 27, 27 |
| $10^{3.1}$ | 2 | 8–13 | 26, 30 |
| $10^{4.1}$ | 2 | 8–9 | 26, 27 |

[a] Based on the relative quantity of LBV RNA, determined by RT-qPCR, in four different areas of the brain.

**Table 3. Evidence for lyssavirus infection, based on different methods, at excretion sites in straw-colored fruit bats with Lagos bat virus-positive brains.**

| | Bat no. | Salivary gland | | | | Tongue | | | | Overall |
|---|---|---|---|---|---|---|---|---|---|---|
| | | RT-qPCR[a] | RTCIT[b] | IHC[c] neuron | IHC epithelium | RT-qPCR | RTCIT | IHC neuron | IHC epithelium (taste bud/ surface) | |
| Naturally infected bat [56] | na | na[d] | na | + | + | na | na | + | + | + |
| Experimentally infected bats | | | | | | | | | | |
| Virus dose group (TCID$_{50}$) | | | | | | | | | | |
| 10$^{0.1}$ | 19 | 39[e] | +[f] | + | -[g] | 36 | + | + | + | + |
| 10$^{1.1}$ | 13 | - | - | + | - | 35 | + | + | - | + |
| | 14 | - | - | - | - | 35 | + | - | - | + |
| 10$^{2.1}$ | 9 | - | - | - | - | 39 | + | + | + | + |
| | 10 | - | - | - | - | 36 | - | + | + | + |
| | 11 | - | - | + | - | - | - | + | + | + |
| | 12 | - | - | - | - | 39 | + | - | - | + |
| 10$^{3.1}$ | 7 | 39 | + | + | - | 33 | - | + | - | + |
| | 8 | - | - | - | - | - | - | + | - | + |
| 10$^{4.1}$ | 1 | - | - | - | - | 33 | - | + | + | + |
| | 4 | - | - | - | - | - | - | - | - | - |

[a] RT-qPCR, reverse transcription quantitative PCR for detection of viral nucleic acid

[b] RTCIT, rabies tissue culture infection test for detection of live virus

[c] IHC, immunohistochemistry for detection of viral antigen

[d] na, not applicable, test not performed because sample not available

[e] Ct value, cycle threshold of RT-qPCR

[f] +, positive cell culture, or cells staining positive with IHC

[g] -, negative for RTCIT, no Ct value, or no cells staining positive with IHC

tongue surfaces, likely because they had bitten their own tongues or the metal cages (biting of the metal cages by rabid bats had been observed). Bat 14 also had a fractured second digit. Bat 12 had enlarged lymph nodes.

On histopathological examination, lesions directly associated with LBV infection were limited to the brain. All 11 rabid bats had diffuse meningo-encephalitis, ranging from mild to moderate in severity. In each case the meningo-encephalitis was characterized by the presence of few to a moderate number of lymphocytes around blood vessels in the meninges and brain parenchyma (perivascular cuffing). Lymphocytes surrounding blood vessels in the meninges and brain parenchyma were three cell layers thick at most. In the brain parenchyma, there was a mild increase in the number of glial cells and there were occasional pyknotic or karyorrhectic cells of undetermined origin (ranging from one to eight per five 40x objective fields). Negri bodies were not observed. None of the nine bats that survived until the end of the experiment had histologic changes associated with meningo-encephalitis.

The naturally-infected bat [56] was not examined for the presence of gross lesions. On histopathological examination, this bat had a moderate diffuse meningo-encephalitis similar to that seen in the experimentally-inoculated bats that became rabid. The naturally-infected bat had inflammation of the salivary gland (sialoadenitis) which was characterized by multifocal aggregates of moderate numbers of lymphocytes within the interstitium surrounding the larger excretory ducts of the salivary gland. No experimentally-infected bat had this lesion and,

as these lymphocytic aggregates were not co-localized with cells containing virus antigen (see below), it was not clear if this lesion was caused by LBV infection. There were several lesions that we considered to be incidental in the tissues of both the experimentally-infected bats and the naturally-infected bat (S1 Text).

## Immunohistochemistry

Cells that stained positively for lyssavirus antigens using immunohistochemistry were found in a number of tissues. In all cell types in which antigen was detected, it was located in the cytoplasm and consisted of variable numbers of small (approximately 2 μm diameter) granules. The majority of antigen-positive cells did not show any signs of degeneration or necrosis; only a few antigen-positive neurons in the brain showed evidence of degeneration, characterized by cell shrinkage and loss of Nissl substance.

Antigen-positive cells were present in the brains of all 11 experimentally-inoculated bats that died with rabies, but in none of the bats that survived to the end of the experiment. In most of the rabid bats, positively-stained cells could be clearly identified as neurons based on their morphology. In addition to neurons, it is possible that some glial cells were antigen-positive.

In extra-encephalic tissues of experimentally-infected bats, antigen-positive cells were detected in the tongue, heart and salivary gland of some of the bats that died of rabies (Table 4). In the LBV-positive tongues, both neuroepithelial cells of taste buds and epithelial cells of the tongue surface were antigen-positive (Fig 3), in addition to neurons within ganglia.

**Table 4. Lyssavirus antigen expression in peripheral nerve ganglia and tongues of rabid bats, based on immunohistochemical analysis.** No antigen was detected in the lung, kidney, liver, spleen, duodenum, jejunum or large intestine of any of these bats.

| | Bat no. | Day of death (dpi) | Detection of lyssavirus antigen by immunohistochemical analysis | | | | |
| --- | --- | --- | --- | --- | --- | --- | --- |
| | | | Ganglia | | | Epithelial cells (tongue) | |
| | | | Heart | Salivary gland | Tongue | Taste bud | Surface |
| Naturally infected bat [56][a] | na[b] | na | na | +[c,d] | + | na | + |
| Experimentally infected bats | | | | | | | |
| Virus dose group (TCID$_{50}$) | | | | | | | |
| $10^{0.1}$ | 19 | 61 | + | + | + | + (2/4)[e] | + |
| $10^{1.1}$ | 13 | 12 | [f] | + | + | na[f] | - |
| | 14 | 7 | na | - | - | na | - |
| $10^{2.1}$ | 9 | 10 | + | - | + | + (3/22) | + |
| | 10 | 13 | + | - | + | + (2/4) | + |
| | 11 | 17 | na | + | + | + (3/8) | + |
| | 12 | 15 | na | - | - | - (0/6) | - |
| $10^{3.1}$ | 7 | 8 | na | + | + | na | - |
| | 8 | 13 | - | - | + | - (0/9) | - |
| $10^{4.1}$ | 1 | 8 | - | - | + | na | + |
| | 4 | 9 | + | - | - | na | - |
| Tropism (pos/total), ratio in% | | | 4/7, 57% | 4/11, 36% | 8/11, 72% | 4/6, 67% | 5/11, 45% |

[a] additionally ganglia in the intestines were positive in this bat

[b] na, not applicable, data not applicable, or cell types or ganglion was not present in the tissue slide

[c] myoepithelial cells of the acini of the mucous salivary gland were positive in this bat

[d] +, antigen present

[e] (x/y), x is number positive, y is number of taste buds present in the tissue slide

[f] -, antigen not present

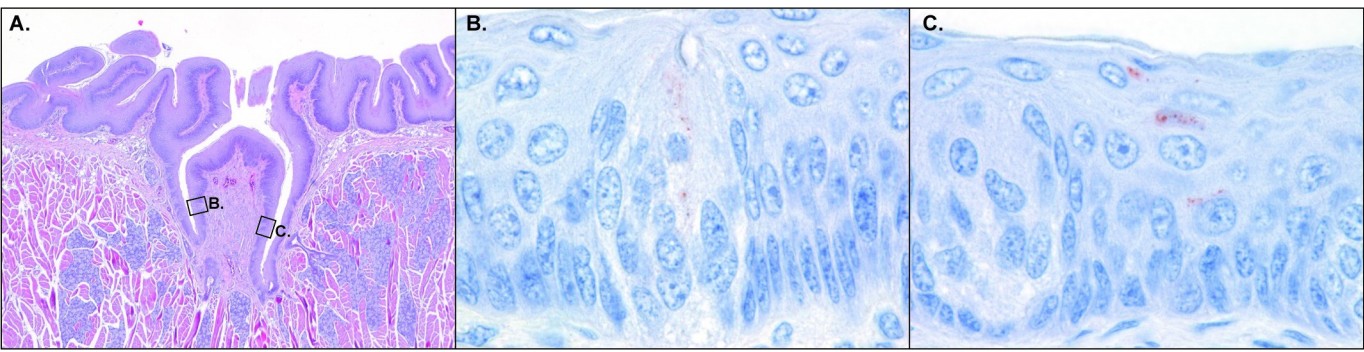

**Fig 3. Microscopy figure of experimentally infected straw-colored fruit bat tongue.** A. Circumvallate papilla characterized by a dome in the surface lined by epithelium. Salivary glands are present in the pit of the dome. HE stain. Original magnification 4x objective. B. Higher magnification of a taste bud that was present in the circumvallate papilla shown in A. The neuroepithelial cells of the taste bud express lyssavirus antigen (red granules). Lyssavirus IHC stain. Original magnification 100x objective. C. Higher magnification of another taste bud that was present in the circumvallate papilla shown in A. Surface epithelial cells near the taste bud express lyssavirus antigen (red granules). Lyssavirus IHC stain. Original magnification 100x objective.

In the LBV-positive salivary glands and hearts, only neurons within ganglia were antigen-positive (Table 4). Bat 19 ($10^{0.1}$ TCID$_{50}$ group), the bat with the longest incubation period, had the highest number of LBV-positive extra-encephalic tissues.

Immunohistochemistry results and RT-qPCR results did not match across the extra-encephalic tissues of the 11 experimentally-inoculated bats that died of rabies. This was especially the case for heart (4 of 7 tested did not match, 57%), tongue (4 of 11, 36%), and salivary gland (3 of 11, [27%] [S2 Table]).

In general, the antigen distribution and staining pattern of the naturally-infected bat was similar to that seen in the experimentally-inoculated bats that died of rabies, with three differences. In the naturally-infected bat, but not in the experimentally-infected bats, antigen-positivity was seen in myoepithelial cells of the mucous salivary gland (S1 Fig), the myenteric ganglia of the intestines, and in the skeletal muscle of the tongue. The antigen-positive structures in tongue muscle of the naturally-infected bat were interpreted to be neuromuscular junctions, because virus antigen distribution was composed of multiple distinct aggregates (S2 Fig), and was not evenly spread throughout the cytoplasm of the muscle fibers, as would be expected if the muscle fibers themselves had been infected.

## Discussion

In this experiment, inoculation of LBV into the masseter muscle caused rabies in 11 of 20 straw-colored fruit bats. The incubation period, duration of clinical disease, and clinical characteristics of the rabid bats were consistent with those described for lyssavirus infections in both bats and carnivores [26,38,45,62]. In our LBV-straw-colored fruit bat model, most bats became rabid within 2 weeks pi. According to our results, a relatively long incubation period seems to be the exception rather than the rule, with few individuals having incubation periods two to five times the mean of that of the others. As in our current study, exceptionally long incubation periods have been previously found to occur in low virus dose groups [26,38,45], the reasons for which are not clear. A possible explanation is that a low initial virus dose results in a small number of initially infected neurons, which consequently lowers the number of neurons infected in the brain in subsequent rounds of interneuronal transmission of virus, and in this way increases the time to development of clinical signs. For both experimental and natural lyssavirus infections, durations of disease of 1–28 days were described for insectivorous and frugivorous bats [33,38,63–66], and of 2–14 days for carnivores [62]. Thus, the duration of

clinical disease seen in our study is within the previously-reported range for bats, but clearly at the lower end. We observed duration of disease of up to 24 hours, but it should be noted that six of the seven bats that developed clinical signs died, or were euthanized, within 12 hours of these signs first being seen. Also, four of the bats that developed rabies were found dead with no observed signs of disease. The clinical signs seen in our bats were similar to those described in our previous experiment with this strain of LBV [55], and are generally similar to those described for rabies in dogs [62].

Our results show that out of five doses of LBV tested via intramuscular inoculation, the middle inoculation dose ($10^{2.1}$ TCID$_{50}$) was most successful in causing infection in straw-colored fruit bats: all four bats inoculated with this dose became rabid, while for each of the lower and higher doses this was maximally two of four inoculated bats (Table 1). The optimal inoculation dose of LBV seemed to follow the so-called "Goldilocks principle": not too high, not too low, but just the right amount [67]. This may appear to be counterintuitive because, for experimental inoculations of most viruses, the rule is that a higher inoculation dose increases the likelihood of infection [68]. Our findings are partly corroborated by those reported by others that used multiple doses of lyssavirus (all RABV) via the intramuscular route. Turmelle et al [45] tested six doses where the middle two caused the highest mortality rates. Franka et al and Baer et al [26,31] tested two different doses and the lowest dose caused the highest mortality rate. In contrast, three other groups found that their highest dose, out of three or four doses tested, caused the highest mortality rate [25,38,69](Fig 4).

One explanation for the survival seen with higher lyssaviral doses in our study is that a high dose increases the chance of triggering an effective immune response [70]. By this, we mean an immune response that stops the virus infection before clinical signs occur and the bat dies. Such an immune response could develop either due to the inoculum itself, or due to a limited infection that has not yet spread widely. The inoculum could contain, besides infectious virus, defective interfering viral particles that could trigger the development of an immune response [71]. We aimed to keep the level of defective interfering particles as low as possible by making a virus stock grown up from a low dose inoculum, and with minimal passage in cell culture. We used the same virus stock diluted to different final concentrations for the different groups. Therefore, we do not expect there to have been a different ratio of infectious virus particles to defective interfering particles in each inoculum. In our experiment, there was a trend for more frequent detection of virus neutralizing antibodies (VNA) against LBV in the serum of survivors from the high dose groups ($10^{3.1}$ and $10^{4.1}$ TCID$_{50}$, Fig 2), although this was not statistically significant. For two other research groups that performed experiments with different doses, VNAs were more often detected in the serum of survivors in high dose groups in one study [45], but not in the other [31]. Further experiments would be needed to determine whether high lyssavirus doses inoculated via a natural route have a vaccination-like effect in bats.

A possible explanation for the survival observed with lower lyssavirus doses, when VNAs did not develop, is that these doses were approaching the minimal infectious dose for this virus via this inoculation route. The fact that, both in our study and those of others, lyssavirus dose appears to follow the Goldilocks principle, indicates the importance of selecting the appropriate virus dose when conducting lyssavirus animal experiments. It is important to take this principle into account when judging the pathogenicity of a virus strain based on mortality rate in experimental studies [72].

In our experiment, it seemed that the majority of bats with VNA showed no clinical signs and survived. When combining the results of previous experiments with bats, using RABV [31,45], there is a positive correlation between survival, and the detection of VNA: of 30 bats that survived inoculation 16 (53%) developed VNA, while of 24 that died, 3 (13%) had

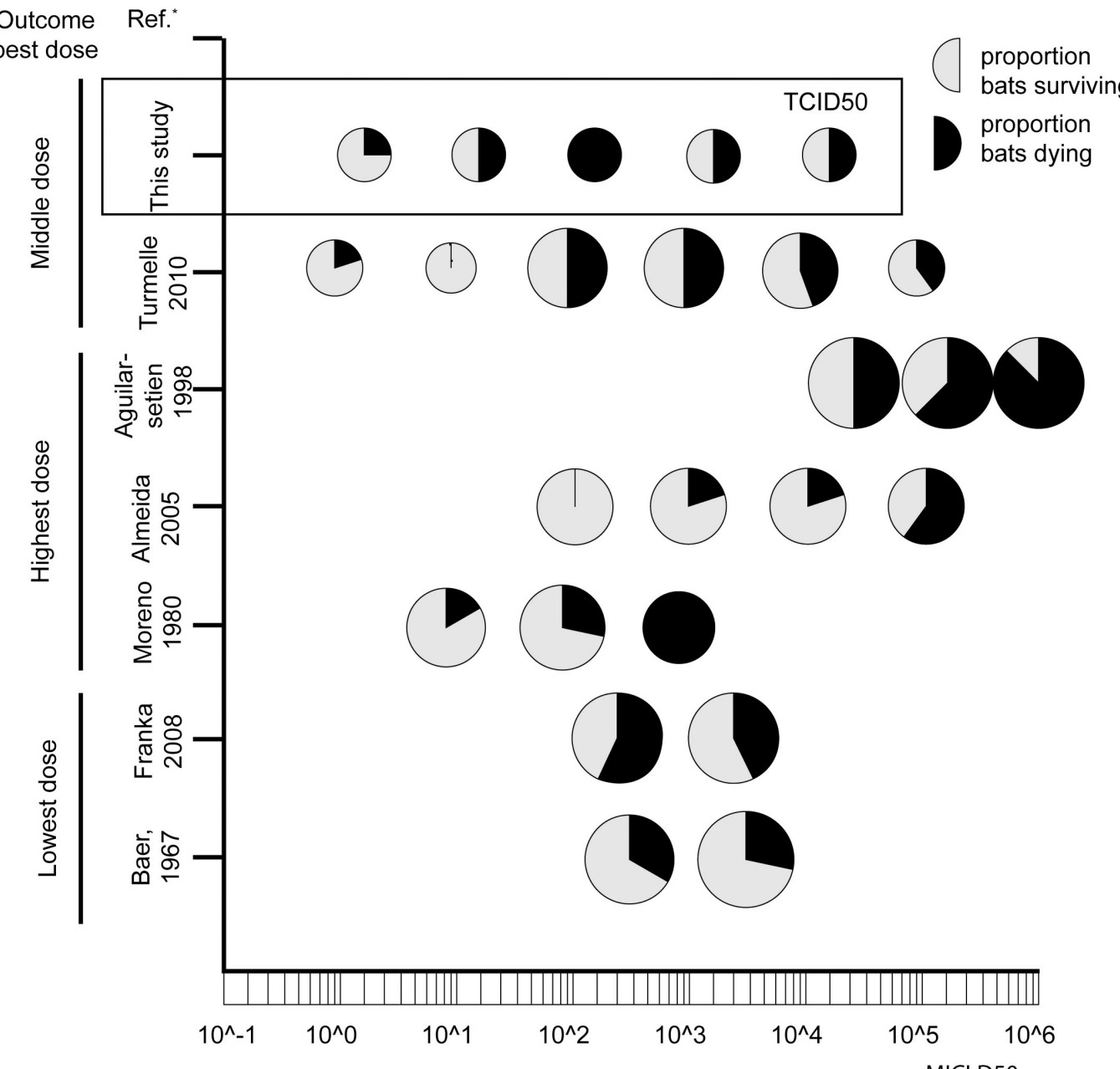

**Fig 4. Schematic overview of mortality and survival of bats in published experiments that used different lyssavirus doses, compared to the results of this study.** Only studies conducted since 1960 and experiments using intramuscular inoculation routes are included. Each circle is a group inoculated with a certain dose. The size of the circle indicates the number of animals included in the group. Black indicates ratio bats in the group that died due to infection, grey means ratio that survived. Published experiments used rabies virus, our study used Lagos bat virus.

developed VNA [$X^2$ (2, $N$ = 54) = 9.7, $p$ = .0018]. In bats that survived RABV inoculation, clinical signs had not been observed, nor was excretion of virus detected. These bats also had no histologic lesions indicative of encephalitis. Therefore, there is no evidence that the virus had

reached the brain before being cleared by the immune response. Thus, our findings and those of others strongly suggest that bats can develop an immune response without undergoing clinical disease from lyssavirus infection, or excreting virus. This could be an explanation for the high seroprevalence detected in free-ranging straw-colored fruit bat populations [54], as well as in some other bat species (Table 3.2 [22])[73]. In populations that have a high seroprevalence, protection against lyssavirus infection is thus high, so prevalence of virus infection is low, and the low level of lyssavirus detection reported can hence be expected [74]. We suggest that high seroprevalence in combination with low virus prevalence might occur when bats in roosts experience lyssavirus exposure in the absence of productive infection. This could be through exposure to high titers of virus (as in our high titer groups) or if multiple exposures to low virus titers occurred and stimulated high antibodies responses.

In our experiment (Fig 2), and that of others using RABV [31,45], a minority of bats with VNA died. An explanation is that in these cases, VNA had developed too late relative to the extent of lyssavirus infection in the CNS. Besides the timing of VNA development, the ability of VNA to cross the blood-brain barrier might be important for the chance of survival. In a study with eight dogs experimentally infected with RABV, those that died only had VNA in the serum, while those that survived also had VNA in the CNS fluid [75]. In that study, dogs that had survived the experiment had lesions in the brain, suggesting RABV had reached the brain before being cleared by the immune response. In bats, the presence of lyssavirus-specific antibodies in the cerebro-spinal fluid has not been investigated.

The method of dissemination of LBV within the straw-colored fruit bat is most likely similar to that of RABV in other mammals, i.e. via synaptically connected neurons. Virus was inoculated into the masseter muscle and was detected in the brains of bats that became rabid. Thus, possible routes of virus spread from the inoculation site (masseter muscle) to the brain (Table 5) can be deduced. A time course experiment would elucidate which of these routes are taken by LBV, and this would help us to understand where virus enters the central nervous system, which would help to explain or predict clinical signs [48]. Likewise, the detection of virus in certain peripheral structures away from the inoculation site can be used to infer which routes of spread the virus might have taken from the brain to these sites (Table 6). Virus detection in the otic ganglion of the salivary gland, and in the ganglion of the tongue indicates LBV spread from the brain to the periphery via parasympathetic motor neurons, which is consistent with the results of our previous experimental infection of straw-colored fruit bats with LBV [55]. Virus detection in the taste buds of the tongue indicates that LBV was also able to spread

**Table 5. Possible neuronal routes for Lagos bat virus to spread from the intramuscular inoculation site to the brain (centripetal spread).**

| Peripheral site of inoculation | Potential routes for centripetal spread of LBVs from intramuscular inoculation site to brain | | | | |
| | Innervation of masseter muscle | | | Characteristics of neuronal route | |
| | Specific nucleus | General location of nucleus in CNS | Route (nerves & ganglia) | Number of synapses to pass | Division of nervous system involved |
|---|---|---|---|---|---|
| Masseter muscle | Motor nucleus of the trigeminal nerve in the brainstem | Motor medulla | Trigeminal nerve (via neuromuscular junction) | 1 | Somatic motor |
| | Rostral sensory mesencephalic nucleus | Sensory medulla | Trigeminal nerve (via neuromuscular spindle) | 1 | Somatic sensory |
| | Motor nucleus of the trigeminal nerve in the brainstem | Motor medulla | Trigeminal nerve | 2 | Parasympathetic motor |
| | Motor nucleus of the trigeminal nerve in the brainstem | Motor medulla | Trigeminal nerve | 2 | Sympathetic motor |
| | Rostral sensory mesencephalic nucleus | Sensory medulla | Trigeminal nerve | 1 | Visceral sensory |

**Table 6. Possible neuronal routes of spread of Lagos bat virus from the brain to the periphery (centrifugal spread).**

| Peripheral location where Lagos bat virus was detected by immunohistochemistry | Potential routes for centrifugal spread of LBVs from brain to periphery | | | | |
| --- | --- | --- | --- | --- | --- |
| | Origin in CNS | | | Characteristics of neuronal route | |
| | Specific nucleus | General location of nucleus in CNS | Route (nerves & ganglia) | Number of synapses to pass | Division of nervous system involved |
| Tongue ganglion | Salivatory | Motor medulla | Chorda tympani and glossopharyngeal nerves | 1 | Parasympathetic motor |
| Otic ganglion (salivary gland) | Inferior salivary | Motor medulla | Chorda tympani and glossopharyngeal nerves | 1 | Parasympathetic motor |
| Cardiac plexi at base of heart[a] | Dorsal motor nucleus of vagus nerve | Motor medulla | Vagus nerve | 1 | Parasympathetic motor |
| | Cranial four to five segments thoracic spinal cord | Spinal cord | Cervical paravertebral sympathetic trunk and postganglionic fibers | 2 | Sympathetic motor |
| Taste buds on tongue | Solitary tract | Sensory medulla | Geniculate, petrosal and nodosal ganglia | 2 | Special sensory |
| Myoepithelium of parotid or submandibular salivary gland[b] | Intermediate horn grey matter of spinal cord, T1 level | Spinal cord | Ventral root of spinal cord, cranial cervical ganglion | 2 | Sympathetic motor |
| | Salivatory | Motor medulla | Parotid gland: cranial nerve IX and otic ganglion; submandibular: cranial nerve VII and submandibular ganglion | 2 | Parasympathetic motor |

[a] More than one route possible because of innervation of the nervous plexi by both parasympathetic and sympathetic nervous systems.

[b] Virus antigen was detected here in the naturally infected bat, not in the experimentally infected bats.

via special sensory neurons, from which it could infect the neuroepithelial cells that comprise the taste bud. A better understanding of the types of neurons and the types of synapses that lyssaviruses use for dissemination in the host will enable us to better predict the sequence of lyssavirus spread within the host's body after exposure at a certain location. This could help in the management and costs of post exposure prophylaxis in people [48].

In general, the virus distribution pattern and lesions in our study were comparable to those of our naturally-infected bat (Table 4) [56]. The virus distribution pattern was most similar to the naturally-infected bat for bats infected with $10^{2.1}$ and $10^{0.1}$ TCID$_{50}$ LBV doses. This indicates that, in addition to successfully resulting in productive infection, the middle dose results in a course of infection that may mimic natural LBV infection in the straw-colored fruit bat. There was one important exception regarding the antigen distribution pattern: salivary gland epithelium was positive in the naturally-infected bat, but not in any of our experimentally-infected bats. In another straw-colored fruit bat naturally infected with LBV, salivary gland epithelium was also positive [74]. Possible reasons for a lack of salivary gland epithelium infection in our experiment include the bats dying or being euthanized before the infection could reach the salivary gland epithelial cells, and an inability of our LBV strain to infect salivary gland epithelial cells. Some authors (e.g., [47,76]) used viruses isolated from the oral cavity instead of from the brain for experimental infection studies; this may ensure that viruses are from a population that is able to infect cells important for excretion [55].

In general, detection of viral RNA and detection of viral antigen (RT-qPCR/ Immunohistochemistry) did not match closely for the diagnosis of LBV infection in extra-cephalic organs (S2 Table). Neither method seemed consistently more sensitive than the other. This is different from our previous experiment in which RT-qPCR seemed more sensitive than immunohistochemistry in extra-cephalic organs, and where these results generally matched those for

tissues in which ganglia were usually present in microscopy slides, such as heart and intestine [55]. The inconsistency between the two different methods might be caused by the relatively abundant localization of the virus in neuronal cell bodies, which are not evenly spread in tissue samples. Hence the detection of virus is dependent on the chance that an infected neuron cell body ends up in the sample examined. This should be taken into account when the extra-cephalic spread of lyssaviruses is studied.

Both salivary gland and tongue are expected to be virus excretion sites in lyssavirus-infected bats [48]. In our straw-colored fruit bat that was naturally infected with LBV [56], as well as another published report [74], salivary gland epithelial cells and tongue epithelial cells were antigen positive. For all but one of our rabid bats, we showed that virus was present in these organs (Table 3) through the detection of viral RNA, the detection of infectious virus by isolation or by the detection of virus antigen. Tongue (10 of 11) tested positive much more commonly than salivary gland (5 of 11) across all techniques. Also, Ct values were generally lower for tongue (mean 35) than for salivary gland (mean 39), suggesting that there was a higher number of virus copies in tongue than in salivary gland tissue. Data from vampire bats (*Desmodus rotundus*) infected with RABV indicate the possibility that the tongue is the major site of virus shedding [77]. Also, lyssavirus was detected in the tongues of experimentally [32] and naturally-infected European bats [78]. On the other hand, Allendorf et al [50] showed salivary glands as being more often RT-hnPCR positive than tongues in bats of different species naturally infected with RABV. Davis et al [79] detected virus in tongue and salivary gland with a similar frequency and at similar Ct values in silver-haired bats (*Lasionycteris noctivagans*) experimentally infected with RABV. Overall, our results suggest that tongue is a more important tissue for virus excretion than salivary gland for LBV in the straw-colored fruit bat. Further studies of natural infection with this virus-host combination are needed to confirm our results.

The finding of virus in the surface epithelium of tongues, especially, suggests virus shedding could have occurred in our bats, at least by the time of death. Still, it was not possible to confirm virus excretion via oral swabs that were taken shortly before, or at, the time of death and when the bats were rabid. This concurs with the results of our previous experiment in which we inoculated straw-colored fruit bats with LBV via the intra-cranial route [55]. Here, also, even though virus was detected in tongue surface epithelium, we were unable to detect virus in oral swabs. Our lack of detection of beta-actin in 14 (14%) of our samples, might suggest RNA extraction failed, but this would explain the lack of detection of lyssavirus in only a minor proportion of our samples. There are several other possible explanations for the lack of detection of viral excretion, as outlined previously by Suu-Ire et al [55]. First, IHC detection of lyssavirus antigen in oral tissues may be more sensitive than RT-hnPCR detection of lyssaviral RNA in oral swabs. Second, although there was virus in tongue surface epithelium, there might have been no excretion into the oral cavity, although this does not fit well with what is known of the pathogenesis of other lyssaviral infections [48]. Third, it might be that virus excretion was intermittent and oral swabs were taken at a time when virus was not being excreted. Intermittent excretion has been proposed to explain alternating positive and negative results of serially collected oral swabs in other experimental lyssavirus infections in bats [26]. However, virus was not detected by RT-hnPCR in any of our oral swabs. Fourth, there may have been loss or degradation of viral RNA in oral swab samples during transport or processing. LBV has been detected in an oral swab of a naturally infected straw-colored fruit bat previously [74], showing that this is technically possible. Because virus could not be detected in oral swabs, time taken post-inoculation until virus excretion, if it occurred, could not be ascertained or linked to the timing of other factors, such as antibody response, or onset of clinical signs.

Overall, our results indicate that the outcomes of infection in straw-colored fruit bats are, in general, not different from those for RABV infection in other species. Hence, people exposed to LBV should be treated. RABV vaccination and PEP are not likely to protect against LBV, so there is a need to develop treatments that do work for exposure to LBV and other phylogroup 2 lyssaviruses [80]. The LBV dose that most commonly leads to infection in straw-colored fruit bats via the intramuscular (masseter muscle) route of inoculation follows the Goldilocks principle. With the middle virus dose, $10^{2.1}$ TCID$_{50}$, nearly the whole course of lyssavirus infection, from site of entry to site of exit, is replicated. Therefore, intramuscular inoculation of $10^{2.1}$ TCID$_{50}$ of the Ghana strain of LBV in straw-colored fruit bats is a promising experimental model to increase our understanding of the dynamics of lyssavirus infections in bats.

## Materials and methods

### Ethics statement (experiment)

Experimental procedures were approved beforehand by the Wildlife Division of the Forestry Commission of Ghana, the Institutional Review Board of Noguchi Memorial Institute for Medical Research, University of Ghana, Legon and the Ethics Committee of the Zoological Society of London, U.K. (license number WLE638).

### Experimental set up

We inoculated five different virus doses, differing by ten-fold steps from $10^{0.1}$ to $10^{4.1}$ median tissue culture infectious dose (TCID$_{50}$), into straw-colored fruit bats in order to find the virus dose leading to the highest rate of infection. The pathogenesis of LBV infection was evaluated by observation for clinical signs, by the testing of oral swabs and tissue samples for virus presence, the sequential taking of blood for antibody detection and by the examination of tissues for lesions. If LBV infection occurred, lesions and virus antigen distribution were compared with those in a naturally-infected bat, to test if the observed experimental infection mimicked a natural infection.

Although the natural routes of infection are not known for LBV in straw-colored fruit bats, we chose intramuscular inoculation because this is the most-commonly used route of inoculation for lyssavirus experiments [48]. The masseter muscle was chosen because it is easily identifiable and because we considered it a muscle likely to be bitten by fighting bats and, hence, a likely natural route of infection. The course of infection in inoculated bats was followed until general paresis was reached in order to provide maximum time for the virus to spread from muscle to brain and peripheral sites, including potential site(s) of excretion, and to determine whether bats were able to survive infection. Thus, virus antigen distribution was examined at the end stage of disease. Oral swabs and blood samples were taken regularly throughout the experiment, to be able to test when bats excrete virus in relation to the time of inoculation, the start of clinical signs, humoral immune response and death.

### Virus preparation (experiment)

A virus stock was prepared and titrated according to standard methods [61]. The virus was from a phylogenetic lineage A of LBV. It was isolated from the brain of a naturally-infected straw-colored fruit bat in Kumasi, Ghana (GH 325, FLI lab. No.: 31225, INSDC sequence databases LN849915, GenBank: LN849915.1 [56]). The virus was passaged three times in baby hamster kidney (BHK) cells. It reached an infectious virus titer of $10^{7.25}$ TCID$_{50}$ per ml. The volume of the inoculation dose per bat was 30 µl. The neat dose was $10^{4.1}$ TCID$_{50}$. For each

subsequent dose, the virus suspension was diluted 10-fold in minimal essential medium (MEM), until a dose of $10^{0.1}$ TCID$_{50}$ was obtained.

## Bats (experiment)

Bats were obtained from a closed captive breeding colony that is maintained in Ghana [81] and which has tested free of LBV [55]. The bats used in this experiment were all captive-bred and each one tested negative for antibodies against LBV using a modified version of the fluorescent antibody virus neutralization test with a lineage A LBV as the challenge virus [54] at the beginning of the study. Age class was based on the approximate date of birth as per the microchip number: all captive bats were caught-up quarterly and any new pups or juveniles were microchipped. Housing and care were as described previously [55]. Twenty bats, all male and 1–2 years old, were randomly assigned to one of five groups (four bats in each group) (S1 Table). The bats in each of the five dose groups were inoculated intramuscularly in the left masseter muscle with 30 μl of one of the five different virus doses. Bats were implanted with transponders to allow individual identification.

## Clinical examination and sampling (experiment)

Bats were initially observed twice daily, at 07.00 and at 16.00 GMT, for the presence of clinical signs. After the first occurrence of clinical signs, frequency of observations was increased to every two hours, day and night. Observers were unaware of the group the bat was in.

Blood samples (0.5 ml) from the propatagial vein were taken at 2 days, 1 week, and 1½ weeks pi, and then at 2, 2½, 3½, 4½, 6½, 8½, 10½, and 12½ weeks pi (day 90 pi). Oral swabs (individually wrapped 2.5 mm diameter sterile cotton tip [Fisher Ltd.]) were taken once daily until 22 days pi, then at days 26, 33, 61, 90, 104, 117, and 146 pi and at the time of euthanasia or at the end of the experiment (day 160 or 161 pi).

Oral swabs (individually wrapped 2.5 mm diameter cotton-tipped, Fisher Scientific Ltd.) were collected in RNAlater (Ambion) and in minimal essential medium, for virus RNA detection and virus isolation, respectively. Swabs were tested for the presence of virus RNA in the following ways: (1) if the bat died during the course of the experiment, all swabs available were tested; (2) if the bat survived until the end of the experiment, only the oral swabs taken at the time of necropsy were tested. The experiment was terminated at day 160 or 161 pi, which was 99 or 100 days after the last bat had died of rabies, at day 61 pi. Bats that survived until day 160 or 161 pi were euthanized and sampled according to the same protocol as clinically affected bats. Bats were euthanized with sodium pentobarbital (0.4 ml/kg body weight, Animal Care Ltd, UK).

## Serology (experiment)

Blood samples were centrifuged (6000 *g*, 15 min; Eppendorf-Netherler-Hinz, GmbH, Germany). Serum was removed with a micropipette. Prior to freezing at -70˚C, sera were heat-treated (56˚C, 30 min) to inactivate pathogens and residual complement. A modified fluorescent antibody virus neutralization test [54] was used for the detection of LBV lineage A specific antibodies. The lack of accurately-titered control sera meant that all neutralizing antibody levels were expressed as reciprocal titers, which were calculated using the Spearman–Karber method. Rabbit sera containing antibodies directed against LBV lineage A were used as positive controls. We did not have access to true specific pathogen free bat sera, and as Lagos bat virus is absent in Europe, we used dog sera from unvaccinated healthy European dogs as negative controls. Bats were considered seropositive if their sera neutralized LBV lineage A at a reciprocal titer higher than 16.

## Pathological examination (experiment)

Necropsies and tissue sampling were performed according to a standard protocol, approximately 1 to 6 hours after death. At necropsy, a standard range of tissues (see below) was collected (1) fixed in neutral-buffered 10% formalin for histological examination, (2) in plain dry tubes for virus isolation and viral RNA detection. The following tissue samples were collected: brain (rostral cerebrum, hippocampus, cerebellum, medulla oblongata in separate tubes for virological analysis, the remainder of brain for histological analysis), salivary gland (usually parotid, but sometimes submandibular), tongue, heart, lung, liver, kidney, spleen, submandibular lymph node, duodenum, jejunum and colon. Each tissue was collected using a new pair of disposable forceps and a new scalpel blade on an individual gauze pad to prevent cross-contamination. For the tissues for histological examination, the formalin was replaced after two or three days to enhance fixation; the formalin-fixed samples were stored at room temperature. The plain samples in dry tubes were flash-frozen at -70˚C.

## Virological examination (experiment)

A range of tests were performed in biosafety level 3 laboratories at the Animal and Plant Health Agency, U.K. and at the Friedrich Loeffler Institute, Germany, as described below:

*Reverse transcription-PCR* To obtain RNA from tissues, a small piece (ca. 50μg) of each organ was homogenized with a 5 mm steel bead in 500 μl MEM with a homogenizer (Tissue Lyser II Qiagen, Hilden, Germany) at 3 minutes for 30 Hz. Homogenates were clarified by centrifugation (500 $g$, 5 min). The clarified supernatant was split and then subject to virus isolation (below) and RNA extraction. For RNA extraction the NucleoSpin RNA kit (Macherey-Nagel GmbH & Co. KG, Düren, Germany) was used according to the manufacturer's instructions and RNA was stored at −80˚C until use. To obtain RNA from oral swabs, oral swabs stored in RNAlater were used. Extraction was performed with High Pure RNA Isolation kit (with poly A), following manufacturer's recommendations. All brain samples from all bats included in the experiment, and extra-encephalic tissues of clinically affected bats, were tested using RT-qPCR, according to previously described methods [82,83] with a probe specific for Lagos bat viruses from lineages A, B and C: [TxRd]AMAAGATTGTTTTCARGGTKCAYAATCA[BHQ2]. All oral swabs of clinically affected bats were tested using RT-qPCR according to Marston et al. [84]. As a positive internal control for RNA extraction, we performed RT-qPCR for beta actin on each extracted sample.

*Fluorescent antibody test* Touch impressions on glass microscope slides of three different parts of frozen brain (hippocampus, cerebellum, medulla oblongata) of all bats were made and stained with fluorescein-isothiocyanate (FITC)-conjugated anti-rabies mouse monoclonal antibody (Fujirebio Diagnostics, USA, anti-N) [61]. A rabies-virus-positive mouse brain was used as a positive control. The brain of an uninfected mouse was used as a negative control.

*Rabies Tissue Culture Infection Test.* The presence of viable virus was confirmed with the rabies tissue culture infection test (RTCIT) essentially as described [85]. Tissue samples were prepared as described (above). Supernatants were added to mouse neuroblastoma cells (Na 42/13, FLI Cat. No.0229) which were then incubated at 37˚C in 5% $CO_2$ in cell culture flasks and control dishes for three days. After three days the control dishes were fixed and checked for lyssavirus presence using the FAT. In case of negative results, supernatant was discarded and cells were passaged as above. Three consecutive passages with negative result ruled out the presence of viable virus in the sample.

## Histology and immunohistochemistry (experiment and natural infection)

The formalin-fixed tissues were embedded in paraffin wax, cut in 4-μm-thick serial sections, routinely stained with hematoxylin and eosin and examined using a light microscope for the

detection of microscopic lesions. Immunohistochemistry was performed for the detection of lyssaviral antigen [86], as described previously [55]. The brain of an LBV-infected straw-colored fruit bat from a previous experiment [55] was included as a positive control. Immunohistochemistry was performed on samples of brain from all bats, and only on other organ samples of those bats that had positive RT-qPCR results for the brain.

## Naturally infected bat

An apparently-healthy free-ranging straw-colored fruit bat was captured near the zoo of Kumasi on 4 May 2011 (GH235 & V15-373) [56]. It was anesthetized and euthanized by exsanguination. The bat was not examined for the presence of gross lesions. A range of tissue samples (brain, lung, intestine, salivary gland [not specified], tongue, kidney, liver, spleen) was fixed in neutral-buffered 10% formalin for histological examination and frozen at –70˚C in plain dry tubes for virus isolation. This bat was part of a larger study to determine virus presence in a colony of straw-colored fruit bats in Kumasi, Ashantia Region, Ghana. The brain of this bat was the only one of 600 (frozen) bat brain samples that tested positive by RT-PCR for the presence of lyssavirus [56]. The Lagos bat virus that was cultured from the brain of this bat was the virus used to inoculate the bats in the current study. Formalin-fixed tissues were processed for the detection of lesions and virus antigen with light microscopy as described above ('Histology and immunohistochemistry'). As tissues had been stored in formalin for over three years, antigen retrieval was increased by boiling slides in citric acid buffer for 20 minutes, in contrast to the 10 minutes for the tissues of the experimental bats. A small part of the brain, at the level of the hippocampus, was available for microscopy.

## Supporting information

**S1 Text. Incidental lesions detected in bats from the experiment and the natural-infected bats.**
(DOCX)

**S1 Table. Age category and body weight of bats inoculated with different doses of Lagos bat virus.**
(XLSX)

**S2 Table. Virus RNA and antigen detection (RT-qPCR/Immunohistochemistry) in extracephalic organs of bats from the experiment do not match well.**
(XLSX)

**S1 Fig. Microscopy figure of naturally infected straw-colored fruit bat salivary gland.** Mucous salivary gland acini separated by septa are shown. One of the lining myoepithelial cells expresses lyssavirus antigen (red granules). Lyssavirus IHC stain. Original magnification 100x objective.
(TIF)

**S2 Fig. Microscopy figure of skeletal muscle in the tongue of a naturally infected straw-colored fruit bat.** Antigen positive granules are interpreted as being localized in neuromuscular junctions, because the positive granules form discrete aggregates rather than being dispersed evenly throughout the cytoplasm of the myocytes. Lyssavirus IHC stain. Original magnification 40x objective.
(TIF)

## Acknowledgments

We thank Ivan Kuzmin for assistance during the experiments and critical review of the manuscript. We thank the management and the staff of the Wildlife Division of the Forestry Commission in Ghana for the care of the bats; the Virology and Animal Experimentation Divisions of the Noguchi Memorial Institute for Medical Research for providing laboratory space for processing bat carcasses, and for freezer storage of samples; Lonneke Leijten and Peter van Run for technical support for processing the formalin-fixed tissues; and David van de Vijver for advice on statistics.

## Author Contributions

**Conceptualization:** Daniel L. Horton, Thijs Kuiken, Anthony R. Fooks, Thomas Müller, James L. N. Wood, Andrew A. Cunningham.

**Formal analysis:** Lineke Begeman, Richard Suu-Ire, Ashley C. Banyard, Elisa Eggerbauer.

**Funding acquisition:** Thijs Kuiken, Anthony R. Fooks, James L. N. Wood, Andrew A. Cunningham.

**Investigation:** Lineke Begeman, Richard Suu-Ire, Elisa Eggerbauer, Conrad M. Freuling, Louise Gibson, Hooman Goharriz, Daisy Jennings, Denise A. Marston, Silke Riesle Sbarbaro, David Selden, Emma L. Wise, Andrew A. Cunningham.

**Methodology:** Daniel L. Horton, Anthony R. Fooks, James L. N. Wood, Andrew A. Cunningham.

**Project administration:** Andrew A. Cunningham.

**Resources:** Richard Suu-Ire, Christian Drosten, Yaa Ntiamoa-Baidu, Anthony R. Fooks, Thomas Müller, Andrew A. Cunningham.

**Supervision:** Thijs Kuiken, Anthony R. Fooks, James L. N. Wood, Andrew A. Cunningham.

**Visualization:** Lineke Begeman, Thijs Kuiken, Andrew A. Cunningham.

**Writing – original draft:** Lineke Begeman, Richard Suu-Ire, Thijs Kuiken, Thomas Müller, Andrew A. Cunningham.

**Writing – review & editing:** Lineke Begeman, Richard Suu-Ire, Ashley C. Banyard, Christian Drosten, Elisa Eggerbauer, Conrad M. Freuling, Louise Gibson, Hooman Goharriz, Daniel L. Horton, Daisy Jennings, Denise A. Marston, Yaa Ntiamoa-Baidu, Silke Riesle Sbarbaro, David Selden, Emma L. Wise, Thijs Kuiken, Anthony R. Fooks, James L. N. Wood, Andrew A. Cunningham.

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
