## [Decision Letter · Decision Letter 0]

21 Aug 2020

Dear Dr. Begeman,

Thank you very much for submitting your manuscript "Experimental Lagos bat virus infection in straw-colored fruit bats: a suitable model for bat rabies in a natural reservoir species" for consideration at PLOS Neglected Tropical Diseases. As with all papers reviewed by the journal, your manuscript was reviewed by members of the editorial board and by three independent reviewers. The reviewers appreciated the attention to an important topic. Based on the reviews, we are likely to accept this manuscript for publication, providing that you modify the manuscript according to the review recommendations. 

This is a very interesting and well written paper that just requires some minor revisions as per the comments of reviewers 1 and 3.

Sincerely,

Tom Hughes

Guest Editor

Stuart Blacksell

Deputy Editor

This is a very interesting and well written paper that just requires some minor revisions as per the comments of reviewers 1 and 3.

Reviewer's Responses to Questions

**Key Review Criteria Required for Acceptance?**

**Methods**

-Are the objectives of the study clearly articulated with a clear testable hypothesis stated?

-Is the study design appropriate to address the stated objectives?

-Is the population clearly described and appropriate for the hypothesis being tested?

-Is the sample size sufficient to ensure adequate power to address the hypothesis being tested?

-Were correct statistical analysis used to support conclusions?

-Are there concerns about ethical or regulatory requirements being met?

Reviewer #1: The methods and approach are appropriate, thoughtful, clearly described, and referenced.

Reviewer #2: see overall comments

Reviewer #3: (No Response)

**Results**

-Does the analysis presented match the analysis plan?

-Are the results clearly and completely presented?

-Are the figures (Tables, Images) of sufficient quality for clarity?

Reviewer #1: The results are clearly presented and interpreted in the context of the background literature available (which they took the initiative to summarize in table form). All tables and figures show key material and are clear.

Reviewer #2: see overall comments

Reviewer #3: (No Response)

**Conclusions**

-Are the conclusions supported by the data presented?

-Are the limitations of analysis clearly described?

-Do the authors discuss how these data can be helpful to advance our understanding of the topic under study?

-Is public health relevance addressed?

Reviewer #1: The conclusions match the results of the study and are discussed in appropriate context to stimulate future research to advance knowledge on the topic of bat rabies (with key impacts for human and animal health).

Reviewer #2: see overall comments

Reviewer #3: (No Response)

**Editorial and Data Presentation Modifications?**

Reviewer #1: The background section referenced key literature in the field, but could use updating to include more recent syntheses and context in the field of bat rabies (e.g. doi: 10.1016/B978-0-12-818705-0.00007-8). Please find some minor suggestions below which may improve clarity.

Reviewer #2: (No Response)

Reviewer #3: (No Response)

**Summary and General Comments**

Reviewer #1: The authors presenting very detailed and interesting work regarding an experimental model of bat lyssavirus infection in a natural abundant and widely distributed frugivorous host that is known to be involved in human-wildlife conflicts including disease, crop damage and bushmeat subsistence hunting, thus with potentially high levels of human contact across their geographic range in sub-Saharan Africa. Given the increasing role of bats in emerging zoonoses, this study is timely and could serve as model for the study of bat lyssavirus and other bat virus systems, in efforts to improve global human and animal health systems. It is an important contribution that advances our understanding of potential natural transmission routes of lyssaviruses among bats. As the authors note, this information is critical for improved public and animal health measures to prevent transmission of lyssaviruses to humans and domesticated animals especially in underserved, resource-limited settings. 

L79-81 – please also consider this recent example of suspected natural transmission in a captive setting however – doi: 10.7589/2019-04-108

L147 – please identify the units for value “27”

L149-150 – please clarify that the VNA were detected from serum samples

L276-279 – it seems the authors might also consider and discuss the possibility of defective interference particles (see PMID: 833940). The middle range doses may have fewer DI particles than higher doses, somewhat consistent with serology?

L296-301 – I think the authors may also consider for discussion the potential for differences in isolate pathology or clinical “phenotype” depending on whether it was sourced from brain versus salivary gland material of the naturally infected host? 

Methods – page 29. Please provide units for the number 15.57. If this is a VNA titer, I wondered whether the testing was really precise enough to estimate to two decimal places (hundredths of unit)? 

S1 Table – if all of the study subjects were captive bred, it may be interesting to see and statistically evaluate continuous host age (years, months) data?

Reviewer #2: Comments

Given the safety perspective or potential risk of human infection from Lagos bat virus where there is no vaccine and chance of contact may be possible, it deems extremely necessary to determine at which time point that bats can shed the virus. This may occur regardless of the clinical status (see below). 

The purpose of the study seems clear with an objective to provide further understanding of lyssavirus dynamics in natural bat host (straw-colored fruit bats; not classified as endangered species). As stated in the discussion, the results may not differ much from what having seen in the naturally or experimentally infected RABV. However, the results may or cannot explain entirely when the time will be or which mechanism(s) that determine virus shedding to other animals or humans which is truly difficult. 

As in RABV related dog or human cases (naturally infected), the time once the virus enters the free nerve ending and disseminating throughout the nervous system is relatively constant (requiring active retrograde transport). Periods of time that is variable are at the bite or inoculation point and during centrifugal spread. Mode of transport is passive. 

One intriguing aspect is that RABV can stay silently despite already disseminating widely in the CNS. This period may be short or as long as month and can be estimated by following the patients from the time that they developed neuropathic pain (causing by sensory ganglionitis which occurs during centrifugal not centripetal spread) but did not develop any of the brain or encephalitic or paralytic symptoms (by axonopathy or myelinopathy). Anterior horn cell dysfunction is evident in furious case but not exhibiting any lower motor neuron weakness (please kindly see reports by Hemachudha/ Ugolini detailing pathogenetic mechanisms, propagation route and time to shed). 

These periods during the propagation paths that can be variable affects when the infected victims shed the virus. Shedding may occur even shortly before clinical symptoms. 

In terms of brain pathology, what may be seen as perivascular cuffing or mild parenchymal infiltration can be absent in survivors and even in diseased victims. This may depend on which pathogenetic mechanisms dominate, depending on the genetic code of virus at certain gene region, immune evasive strategy, modification of virus once inside the CNS and bottlenecked by immunological pressure and types of neurons (at spinal cord, at brainstem, at hemisphere) that virus has trans-neuronal spreading. 

By having no pathology and no symptoms (remained healthy) in the group of low titers may not be concluded that the virus does not enter the brain and is neutralized at the inoculation site? (the immune response at the muscle seems scant as shown in the case of RABV). One of eight in the low dose seroconverted whereas other 4 did not. 

It is exciting to explore further whether there is (are) other arms of immune effectors that can eliminate virus in time without causing damage in the brain sufficient enough to see clinical symptoms. This also happened in high dose groups, 4 of 8 had clinical symptoms and the remaining are healthy and seroconverted. Such phenomenon reflects natural resistance not only to low dose (as in Amazonians or wild life capture eskimo) but also to high dose. 

The medium dose only that reliably causes disease in all seems an acceptable protocol, however, non-explainable. 

In conclusion, please see the reviewer as a friendly reviewer who sees this lyssa issue as public health threat and wishes to exchange ideas as experienced in the case of RABV and not to reject nor deny any interpretation. 

What are necessary in the view of clinicians are 

- to determine whether the survivors can still shed the virus or not.

- the state of bats may or may not affect their natural resistance to infection as seen in Nipah, such as during parturition, peripartum, during flirting or mating, at the time of drought, health status and age/sex of bats that are vulnerable to abortive infection.

- Such abortive infection bats that are healthy; will they be able to secrete the virus continuously or intermittently and which factor in controlling it?

These may not be answered by experiments in the laboratory but by surveillance in the field determining the prevalence (sero- or cell mediated responses) and whether the rate of prevalence falls into seasonal pattern and correlated or not with shedding. 

The value of the study by this group as in this study and more in the future will be very useful with the clear objectives of which control the shedding and what factors controlling survival and direct them to become carriers or not. Any specific time of season or any influence from ecosystem, etc.

Reviewer #3: Thank you for the opportunity to review this manuscript. It was an interesting and well written study to read. Lagos bat is a uniquely African lyssaviruses, with frugivorous bats considered the reservoir. As with other bat associated lyssaviruses, we understand very little about the pathogenesis and epidemiology. Experimental infections in the proposed natural reservoir is needed to obtain more information. This is however challenging studies that can also only be performed by a selected few groups globally and animal groups are usually small due to logistical and ethical considerations. Understanding the influence of viral infectious dose on disease progression/or not and development of antibody responses is an important question to answer. This study specifically look at this. 

I have a few specific suggestions/concerns detailed below:

1. In general the authors use very outdated references especially in the introduction of the manuscript. This must be improved and replace with more up to date references. 

Eg. Ref 1 is a 2017 paper. More lyssavirus species have been identified in recent years. Rather use ICTV official reference. Hu et al., 2018; Amarasinghe et al., 2019

Also specifically revise ref 3-6, 17, 8, 47 to name a few. 

Two recent reviews have been published that can also be considered:

Markotter et al., 202Bat-borne viruses in Africa…. doi:10.1111/jzo.12769

Markotter W and Coertse J. Bat lyssaviruses. OIE technical report. Rev. Sci. Tech. Off. Int. Epiz., 2018, 37 (2), 385-400

2. I suggest the authors introduce Lagos bat at the start of the introduction and where it fits into lyssavirus taxonomy and epidemiology

3. Line 98: I do not agree that LBV will be a major public health threat. Yes we may miss/see a few sporadic cases but not “major threat”. Please revise

4. Line 133 and Supplementary S1 refer to age of the bats and a distinction is made between adult and sub adult. How was this distinction determined? The method of defining these age classes may influence the lack in correlation between age classes and infections status. 

5. Line 175 and methodology: The authors state that all oral swabs tested negative. This is very unexpected since virus was detected in the tongue. Did the authors perform additional controls to determine if extractions of nucleic acids from the swabs were successful? 

Furthermore although swabs were collected regularly in the study, the authors decide to only test swabs of bats that survived at the end of the study (not all swabs were tested). This is a major limitation and I suggest the authors include these results. See also point 8. On page 20 the authors also mention that the unexpected negative swab results may be due to technical issues and they refer to a previous paper. I suggest summarizing the issues here also since these negative results are unexpected. 

6. Line 261 – Expand on reasons why a longer incubation period is associated with lower viral doses.

7. Line 277 – The authors refer to the “Goldilocks” principle several times in the publication and this was also previously published. However not all readers will understand this context and I suggest the authors rather just explain the principle. 

8. Line 308-314 need to be revised. On the one hand the authors discussed that their study and others found that bats develop an immune response without clinical diseases and this occur without any viral excretion of the virus. This is the explanation for high seroprevelance observed. However immune response must be in response to an infection. This is contradictory. How can high seroprevalence be observed, if no excretion and very low prevalence of virus? How can a very low prevalence of virus and no excretion in most infections cause such high seroprevalence in bat populations that are known to be thousands and even hundreds of thousands large? See point 5 again where I mentioned that the authors should have tested all swabs collected at different time points, also in the bats that survived, to ensure there was no viral excretion at any time point. 

9. Line 338-340. It is unclear how the results in this study directly relate to the management of human exposures to animals? The authors must please clarify this clearly what they practically mean by this statement. Since the correct management of post exposure decisions is crucial it should not be left to interpretation by the reader. 

10. Methodology: 

Virus preparation (page 22). Pleas provide more detail on the virus used e.g. specific lab number and Genbank accession number for sequence if available. 

Serology: Why did the authors use dog sera and not negative bat sera as a negative control. Non-specific reactions are known to be species specific. 

It is also not clear how the cut-off of 15.57 was determined for sero positivity. Please explain.

PLOS authors have the option to publish the peer review history of their article (what does this mean?). If published, this will include your full peer review and any attached files.

Reviewer #1: No

Reviewer #2: Yes: Thiravat Hemachudha, MD, FACP

Reviewer #3: No
---

## [Editor Report · Decision Letter 1]

16 Oct 2020

Dear Dr Begeman,

Thank you very much for submitting your revised manuscript "Experimental Lagos bat virus infection in straw-colored fruit bats: a suitable model for bat rabies in a natural reservoir species" for consideration at PLOS Neglected Tropical Diseases. 

Thank you for your clear responses to the reviewers. You have addressed all the reviewers concerns and questions and the non-technical summary is clear, concise and accessible to a non-technical audience.

However, there are three issues raised by the reviewers that you have addressed in your response to the reviewers but not in the manuscript. Can you please address these 3 points below in the manuscript.

Thank you.

1) REVIEWER: L276-279 – it seems the authors might also consider and discuss the possibility of defective interference particles (see PMID: 833940). The middle range doses may have fewer DI particles than higher doses, somewhat consistent with serology?

REPLY: We aimed to keep the level of DI particles as low as possible by making a virus stock grown up from a low dose inoculum and with minimal passage in cell culture. We used the same virus stock 

(diluted to different final concentrations) for the different groups, so do not expect a different ratio of infectious virus particles to DI particles.

2) REVIEWER Line 133 and Supplementary S1 refer to age of the bats and a distinction is made between adult and sub adult. How was this distinction determined? The method of defining these age 

classes may influence the lack in correlation between age classes and infections status.

REPLY: Age class was based on the microchip number and approximate date of birth. Unfortunately, due to logistical constraints on the frequency (approx. quarterly) of catching-up all of the 

captive bats and microchipping pups/juveniles, it is not possible to know their exact date of birth.

3) REVIEWER: Serology: Why did the authors use dog sera and not negative bat sera as a negative control. Non-specific reactions are known to be species specific.

REPLY: Although the bats we used were captive-bred in a closed colony in which no bat has tested positive for LBV infection, we did not have access to true SPF bats. Given the absence of Lagos bat 

virus in Europe, we considered the use of European-sourced dog sera as a negative control to be reasonable.

[1] A letter containing a detailed description of the changes you have made in the manuscript. 

Important additional instructions are given below. 

Sincerely,

Tom Hughes

Guest Editor

Stuart Blacksell

Deputy Editor
---

## [Editor Report · Decision Letter 2]

19 Oct 2020

Dear Dr Begeman,

We are pleased to inform you that your manuscript 'Experimental Lagos bat virus infection in straw-colored fruit bats: a suitable model for bat rabies in a natural reservoir species' has been provisionally accepted for publication in PLOS Neglected Tropical Diseases.

Best regards,

Tom Hughes

Guest Editor

Stuart Blacksell

Deputy Editor

---

## [Editor Report · Acceptance letter]

3 Dec 2020

Dear Dr. Cunningham,

We are delighted to inform you that your manuscript, "Experimental Lagos bat virus infection in straw-colored fruit bats: a suitable model for bat rabies in a natural reservoir species," has been formally accepted for publication in PLOS Neglected Tropical Diseases.

Best regards,

Shaden Kamhawi

co-Editor-in-Chief

Paul Brindley

co-Editor-in-Chief
